# Are Parkinson’s Disease Patients the Ideal Preclinical Population for Alzheimer’s Disease Therapeutics?

**DOI:** 10.3390/jpm11090834

**Published:** 2021-08-25

**Authors:** Thomas F. Tropea, Alice Chen-Plotkin

**Affiliations:** Department of Neurology, Perelman School of Medicine, University of Pennsylvania, Philadelphia, PA 19104, USA; thomas.tropea@pennmedicine.upenn.edu

**Keywords:** Parkinson’s disease, Alzheimer’s disease, clinical trial, precision medicine

## Abstract

Concomitant neuropathological hallmarks of Alzheimer’s Disease (AD) are common in the brains of people with Parkinson’s disease (PD). Furthermore, AD biomarkers are associated with cognitive decline and dementia in PD patients during life. Here, we highlight the considerable overlap between AD and PD, emphasizing neuropathological, biomarker, and mechanistic studies. We suggest that precision medicine approaches may successfully identify PD patients most likely to develop concomitant AD. The ability to identify PD patients at high risk for future concomitant AD in turn provides an ideal cohort for trials of AD-directed therapies in PD patients, aimed at delaying or preventing cognitive symptoms.

The symptoms of Alzheimer’s disease (AD) and Parkinson’s disease (PD), the two most common neurodegenerative diseases, present a decade or more after the disease process takes hold. Neuroprotective clinical trials in AD increasingly target early or at-risk groups to prevent or delay the onset of disease, yet therapies that clearly impact the cognitive course remain elusive. Identifying the ideal group of people to target in AD neuroprotective studies remains of key importance.

PD affects over 6 million people worldwide, or 1–3% of people over age 65 [1,2,3], already making it the second most common neurodegenerative disease, with numbers that are growing [4,5]. Bradykinesia plus tremor or rigidity make up the cardinal symptoms of PD [6], although mood, cognition, sleep, and autonomic function are also often affected [7]. Dementia, one of the most devastating complications in PD, is associated with worse outcomes and increased mortality [8,9]. People with PD develop dementia at a higher rate than age-matched peers without PD [10]. Specifically, PD dementia (PDD) affects as many as 83% of PD patients long-term [11] and is typically preceded by a prodromal cognitive state of mild cognitive impairment (PD-MCI) [12]. Neuropathologically, PD is characterized by neuronal inclusions composed of misfolded alpha-synuclein (aSyn) that exist in Lewy bodies [13]. However, PD neuropathology does not exist in isolation; as many as 70% of postmortem brain samples from people diagnosed with PD in life have a secondary neuropathological diagnosis of AD [14,15], defining PD as an AD risk state. 

PD differs from AD in its clinical and neuropathological characteristics. AD is an insidiously progressive cognitive disorder and the most common cause of dementia, affecting an estimated 50 million people worldwide [16]. In the preclinical stage, neuroimaging and molecular changes associated with AD are observed, albeit without clinical signs or symptoms of cognitive impairment or dementia. The prodromal phase is associated with changes in cognitive function without functional or social impairment [17]. The preclinical changes and prodromal phase can begin as many as 20 years prior to the onset of dementia. Dementia secondary to AD typically begins in the 7th decade of life and is characterized by impairments in memory, language, problem-solving, and other domains of cognition [18]. Postmortem examination of AD cases shows significant cortical and medial temporal lobe atrophy, and the neuropathological diagnosis is established by the presence of plaques containing aggregated amyloid-β_1-42_ (Aβ) peptides and neurofibrillary tangles of hyper-phosphorylated tau [19]. 

To date, clinical trials studying compounds aimed at slowing or reversing the course of established MCI or dementia secondary to AD have largely been underwhelming. One of the reasons for these disappointing results might be that the clinical syndrome appears near the end of the pathological cascade [20]. Most studies have targeted groups with MCI defined by genetic, neuroimaging, and biomarker characteristics to have early AD pathology for enrollment in AD neuroprotection trials (Table 1). Yet, even this approach may miss a critically early time point in AD pathogenesis, beyond which interventions will have minimal clinical impact. How, then, might we identify individuals at an earlier stage—a cognitively normal cohort with *incipient* AD pathology—without casting such a wide net as to be impractical, if not altogether infeasible? 

We argue here that individuals with early PD may be exactly the cognitively normal, high-AD-risk population in which interventions are likely to impact cognitive course in a clinically meaningful way. In making our case, we review the considerable overlap between AD and PD, emphasizing neuropathological, biomarker, and mechanistic studies. We then highlight precision medicine approaches to identify people with PD at highest risk of AD, in order to support the feasibility of viewing PD as an ideal preclinical cohort to target AD neuropathology in disease-modifying clinical trials. 

## 1. AD Pathology Is Common in PD Brains and Is Associated with Worse Cognitive Performance during Life

The neuropathological hallmark of PD is the aSyn-containing neuronal Lewy body inclusion. However, co-occurring AD pathology is common among all Lewy body disorder cases, which includes PD, PD with dementia (PDD), and dementia with Lewy bodies (DLB, Figure 1), although exact figures differ between studies. Among published cases with a primary neuropathological Lewy body disorder diagnosis, nearly all have some amount of concomitant tau pathology, with one third of them showing a moderate to severe degree of tau pathology. Roughly 50–70% demonstrated sufficient concomitant Aβ plaques and tau neurofibrillary tangles to warrant a secondary neuropathological diagnosis of AD [14,15]. Moreover, the severity of AD pathology among different brain regions is proportional to the aSyn burden in those regions [40]. Furthermore, tau and aSyn co-aggregate in the same neuronal populations in the amygdala and entorhinal cortex and lesser in the prefrontal cortex [41]. Thus, human neuropathological studies suggest synergy between aSyn, tau, and Aβ with some regional and cellular specificity.

In vivo positron emission tomography (PET) neuroimaging with amyloid specific tracers have helped to describe amyloid pathology at different stages in living PD patients. In early, untreated, cognitively normal PD cases from the Parkinson’s Progression Markers Initiative (PPMI), cerebral amyloid [^18^F] Florbetaben uptake is present in ~20% of cases [43], similar to neurologically normal published cohorts at the same age [44]. Throughout the course of PD, amyloid positivity increases as cognition declines. Indeed, Pittsburgh Compound B (PiB) positivity indicating amyloid deposition is at its lowest in PD with mild cognitive impairment (~5%) and higher in PD cases with dementia (~34%). In cases with DLB, with diffuse neocortical aSyn pathology early in the disease, PiB positivity is at its highest (68%) [45]. Although variability exists between amyloid PET tracers, amyloid appears to accumulate as PD progresses, following patterns of aSyn pathology. Pathological tau PET imaging studies have been more challenging due to off-target binding of available tracers. However, retention of the 3R/4R tau tracer ^18^F-flortaucepir in Lewy body disease cases is intermediate between healthy controls and AD, is higher in temporal-parietal regions in cases with higher cerebrospinal fluid (CSF) amyloid levels, and is associated with higher CSF tau levels and a higher severity of neuropathological tau [40].

The location and severity of aSyn pathology associates with clinical features that patients exhibit during life. That is, PD patients with aSyn pathology found not only in the brainstem but also throughout the limbic system and cortex are more likely to have cognitive impairment than PD patients with less extensive aSyn pathology [46]. People with PD and concomitant AD have more severe motor dysfunction, a higher burden of depression, faster rate of cognitive progression, shorter interval from motor to cognitive symptom onset, impaired language performance, higher rate of nursing home admittance, and higher mortality risk, compared to PD patients without AD pathology [46,47,48]. Specifically, temporal lobe tau burden has been independently associated with antemortem deficits in confrontation naming [40,49]. The combination of aSyn and AD copathology confers a worse prognosis associated with worse cognitive function and higher mortality risk. 

## 2. AD Associated Biomarkers of Neurodegeneration, Tau, and Alpha-Synuclein Associate with Cognitive Performance in PD Cohorts

To obtain a glimpse of the underlying neuropathological process, in vivo biomarker studies are important tools, as they can be obtained from biofluids during life, and patients can be observed after the biofluids have been collected. This approach has been informative in AD, through the development and standardization of CSF and plasma-based biomarkers (Aβ, total tau, phosphorylated tau, and neurofilament light [NFL]) [50]. These biochemical biomarkers are highly specific for underlying axonal degeneration (t-tau and NFL) [51,52,53], Aβ-containing plaques (Aβ) [54], and NFT pathology (p-tau) [55]. Indeed, diagnostic criteria employing these biomarkers have been proposed in the AD field [56], and clinical trials in AD use CSF-based biomarkers as entry criteria [57]. 

Numerous studies have examined AD biomarkers as predictors of dementia in PD. For example, lower baseline CSF Aβ level was shown to predict a faster rate of cognitive decline in a study of 45 cognitively normal PD patients. When compared to subjects above a cutoff value of 192 pg/mL, those with lower Aβ levels had a greater annual decline by 5.85 points on the Mattis Dementia Rating Scale-2 (DRS) [58]. Lower CSF Aβ was also associated with a higher risk of cognitive impairment within 3 years of disease duration in the PPMI cohort [59]. Unlike Aβ, CSF t-tau is not associated with cognitive outcome in PD, while phospho-tau results have been mixed, with some studies showing association with cognitive impairment and others not demonstrating such a relationship [58,60]. Although NFL is not specific to AD, higher plasma NFL levels are associated with cognitive impairment in PD [61]. Beyond biochemical biomarkers, the *APOE* E4 allele remains the strongest genetic risk factor for late onset AD. In PD, carrying one or two *APOE* E4 alleles is also associated with an increased risk for dementia in PD and a faster rate of cognitive decline [62,63]. Furthermore, structural MRI correlates of AD captured in the Spatial Pattern of Abnormality for Recognition of Early Alzheimer’s (SPARE-AD) index associate with cognitive impairment and predict a faster rate of cognitive decline in PD [62]. Thus, cognitive impairment in PD associates with biomarkers of underlying Aβ pathology and axonal degeneration as well as genetic risk of AD, suggesting that AD-related pathophysiology is at least partially causal for the cognitive decline that occurs in the majority of individuals with PD. 

## 3. In Vitro, Cell-Based, and Animal Models Provide Evidence for AD Pathogenic Mechanisms in PD

In vitro studies have long suggested synergy between the key pathological proteins implicated in AD and PD, especially tau and aSyn (reviewed in [64,65]). In particular, Jensen et al. reported over 20 years ago that tau and aSyn can physically interact through pulldowns in human brain lysates [66]. Subsequently, Giasson et al. demonstrated that aSyn induced the fibrillization of tau in vitro and that co-incubation of tau and aSyn accelerated the fibrillization of both proteins [67].

More recently, the discovery that pathological forms of both tau and aSyn may template the misfolding of non-pathological tau and aSyn and that these pathological tau and aSyn species may then propagate from cell to cell has led to new data supporting synergy between AD and PD pathogenic processes in cellular and animal models [68]. For example, Bassil et al. recently showed that co-inoculation of pathological conformations of aSyn and tau into mouse brain increased the formation of tau aggregates, and the absence of endogenous aSyn reduced the formation and spread of tau aggregates [69]. 

Thus, in vitro, cell-based, and animal models support the premise that the presence of aSyn may accelerate the development and spread of at least tau, and possibly AD, pathology.

## 4. Precision Medicine Approaches Can Enrich for Those PD Individuals Most Likely to Develop Concomitant AD Pathology

As summarized in the preceding sections, PD individuals who develop cognitive impairment are more likely to carry the *APOE* E4 genotype, more likely to have low CSF Aβ and high CSF and plasma NFL, and more likely to have positive amyloid PET scans. If we turn these associations on their heads and ask, instead, whether the presence of these AD biomarkers can enrich or identify those PD individuals who are most likely to have concomitant AD pathology, the answer that is emerging is very promising. 

Specifically, in a neuropathological study of 208 LBD cases from Penn, structured as discovery and replication cohorts, with validation in an additional 70 LBD cases from 20 centers in the National Alzheimer’s Coordinating Center (NACC) database, we have shown that genotypes at just three single nucleotide polymorphisms (SNPs), along with age at LBD onset, can be used to calculate a risk score for concomitant AD pathology. PD individuals with AD risk scores in the highest quintile, in turn, were fourfold more likely to have concomitant AD pathology than those in the lowest two quintiles. Importantly, the absolute rate of concomitant AD pathology ranged from 60% to 80% in the highest quintile of AD risk among Penn LBD cases [42]. Put simply, this study suggests that a blood test obtained at any time in PD disease course may be able to identify a sizeable group of individuals with 60–80% chance of having concomitant AD at death. If we infer, based on rates of amyloid positivity by PET imaging in newly diagnosed PD cohorts or PD individuals with minimal cognitive symptoms, that only a small minority of PD individuals have incipient amyloid pathology at these early stages, there is great potential to identify very high-risk PD individuals who are not yet amyloid positive. 

While fourfold enrichment for PD individuals destined to have concomitant AD pathology at death is promising, this may still fall short of the levels of certainty needed to enroll a cognitively normal PD group in higher-risk trials aimed at targeting AD-related pathogenic mechanisms. However, a strategy in which (1) minimally invasive blood draws are used to perform genetics-based risk calculation, enriching for a cohort in which (2) AD biochemical biomarker levels (from the CSF or, increasingly, from the plasma) may further hone accuracy, yielding a subgroup in which (3) PET imaging is used to detect the earliest phases of amyloid deposition, is already feasible. Such a strategy is also likely to yield a sizeable group of PD individuals at high enough risk for AD pathology to warrant that targeted intervention. 

## 5. Concluding Remarks

We close with a few observations that may further strengthen the case for viewing “precision-medicine-identified” PD individuals as an ideal preclinical cohort for AD-directed therapies. First, in contrast to current strategies following high-genetic-risk groups in the general population (e.g., carriers of *APOE* E4 alleles), timelines are compressed, and a starting point for thinking about intervention—the time of PD diagnosis—is clearly indicated. Second, we believe that for individuals who already have a neurodegenerative disease diagnosis (PD), willingness to accept the risks inherent in any experimental therapeutic may differ from those with no neurological signs or symptoms. Finally, compared to individuals with established AD (including, for example, those who would fall under the wide-ranging use cases for the recently FDA-approved amyloid-targeting drug aducanumab), cognitively normal PD patients stand to benefit enormously from arresting the course of cognitive decline.

## Figures and Tables

**Figure 1 jpm-11-00834-f001:**
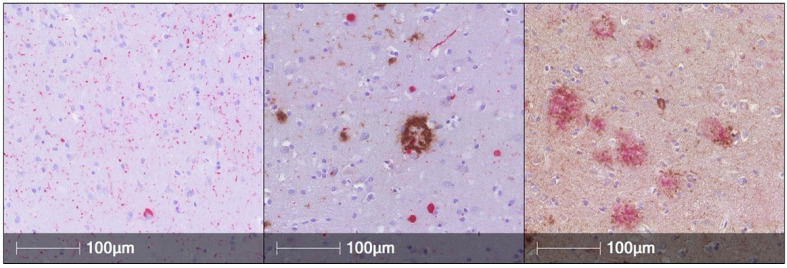
Immunohistochemical sections (160×) demonstrating Lewy Body aSyn (red) pathology in the anterior cingulate cortex (**Left**), concomitant Aβ (brown) and aSyn (red) in the anterior cingulate cortex (**middle**), and Aβ (red) and tau neurofibrillary tangles (brown) in the middle frontal cortex (**Right**). Reprinted with Permission from Dai et al, 2020 [42].

**Table 1 jpm-11-00834-t001:** Alzheimer’s Disease Clinical Trials in Preclinical or Prodromal Participants.

Study Drug	Mechanism of Action	Sponsor	Enrollment Criteria	Ref
Studies enrolling at-risk or preclinical stage human participants				
Atabecestat	BACE Inhibitor	Janssen	*APOE* E4 genotype.	[21]
Celecoxib	Selective COX-2 inhibitor	Pfizer	Cognitively normal with a family history of AD.	[22]
Crenezumab	Aβ monoclonal antibody	Hoffmann-La Roche	*PSEN1* E280A mutation carriers.	[23,24]
Gantenerumab	Aβ monoclonal antibody	Hoffmann-La Roche	*APP*, *presenilin-1*, or *presenilin-2* carriers.	[25]
Simvastatin	HMG-CoA reductase inhibitor	Merck	Cognitively normal with a family history of AD.	[26]
Solenezumab	Aβ monoclonal antibody	Eli Lilly	*APP*, *presenilin-1*, or *presenilin-2* carriers.	[25]
Studies enrolling prodromal human participants				
Aducanumab	Aβ monoclonal antibody	Biogen	MCI with positive amyloid PET.	[27]
Atabecestat	BACE Inhibitor	Janssen	MCI with pathological CSF Aβ or positive amyloid PET.	[28]
BI 409306	Phosphodiesterase-9A inhibitor	Boehringer Ingelheim	MCI.	[29]
Crenezumab	Aβ monoclonal antibody	Hoffmann-La Roche	Pathological CSF Aβ or positive amyloid PET.	[23,24]
Donanemab	Aβ monoclonal antibody	Eli Lilly	MCI with positive amyloid PET.	[30]
Elenbecestat	BACE inhibitor	Biogen, Eisai	MCI.	[31]
Exenatide	Glucagon-like peptide-1 agonist	Astra Zeneca	MCI.	NA
Gantenerumab	Aβ monoclonal antibody	Hoffmann-La Roche	MCI with pathological CSF Aβ.	[25,32]
JNJ-63733657	Tau monoclonal antibody	Janssen	Subjective cognitive decline and positive tau PET.	[33]
Pepinemab	Semaphorin 4D monoclonal antibody	Vaccinex	MCI with pathological CSF Aβ or positive amyloid PET.	[34]
Semorinemab	Tau monocloncal antibody	Genentech	MCI with pathological CSF Aβ or positive amyloid PET.	[35]
Simvastatin	HMG-CoA reductase inhibitor	Merck	MCI.	[36]
Solenezumab	Aβ monoclonal antibody	Eli Lilly	MCI with positive amyloid PET.	[37,38]
Verubecestat	BACE inhibitor	Merck	MCI with positive amyloid PET.	[39]

BACE = β-site amyloid precursor protein cleaving enzyme. Aβ = amyloid- β. MCI = mild cognitive impairment. PET = positron emission tomography.

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
