# Peer review of "Are Parkinson’s Disease Patients the Ideal Preclinical Population for Alzheimer’s Disease Therapeutics?"

_jpm, 2021, doi:10.3390/jpm11090834_

Round 1
Reviewer 1 Report
In the present review, Tropea and Chen-Plotkin raise the question whether Parkinson’s disease (PD) patients would be the ideal preclinical population to test Alzheimer’s disease (AD) medication. In other words, they propose that individuals with PD may be even better suited to evaluate the efficiency of these drugs then AD patients themselves. After giving a brief introduction providing background information on both diseases, the authors list features shared between both neurodegenerative disorders and point out that PD patients are at a very high risk to develop concomitant AD. To undermine their claim, the authors summarize several neuropathological, mechanistic and biomarker studies. In the end, they conclude that, as hypothesized in the title, individuals with PD identified with a high risk for AD probably would majorly benefit from receiving early AD treatment and therefore, would also be an easily assessable cohort to evaluate AD-directed medications.
Broad comments
First of all, the review perfectly fits into the scope of the “Journal of Personalized Medicine”, particularly in the special issue “Personalized Medicine for Parkinson's Disease: New Concepts and Future of Individualized Management”, as it discusses how precision medicine can help to identify PD patients at risk for developing concomitant AD and how this population simultaneously could be a well-suited cohort to test AD medications. Secondly, the manuscript is well-structured, well-written and provides concise background information about both neurodegenerative disorders, PD and AD, thus making it easy for the reader to follow the proposed concept. Moreover, the authors clearly lay out the arguments in support of their hypothesis, which are backed up by findings in recent studies. There are only some minor weak points to consider which are listed in the following.
- The wording of the title (“Is PD the ideal preclinical population for AD therapeutics”) is not quite correct as PD itself can’t be a population. Therefore, it would be more precise to write “Are PD patients the ideal preclinical population for AD therapeutics?”
- In the Table, clinical trials testing various AD-directed drugs are listed. It would be interesting to know whether patients with PD diagnosis were excluded from these study cohorts.
- Minor language recommendations are listed in more detail in the specific comments
Specific comments
1) Language:
- Line 23: insert “the” before “cognitive course”
- Line 29: add “also” before “often”
- Line 56: emphasize this important point by writing“One of the reasons for these disappointing results might be that …”
- Line 137: insert “for” before “dementia”
- Line 174: replace “that” with “than”
- Line 192: it would be clearer to write “…pathology to warrant targeted intervention”
2) Abbreviations:
- Line 98: the abbreviation CSF appears here for the first time, however, it is introduced as late as in line 118
- Line 192: please explain the abbreviation “DRS”
- Line 164: write “NFL” instead of “NfL” to be consistent with nomenclature
- Line 170: please explain “NACC”
3) Figure Legend:
Line 214: add (“right”) after “middle frontal cortex” to be precise about what is shown in the figure
Author Response
Response to Reviewers re: Manuscript ID: jpm-1305508:
We would like to thank the reviewer for their thoughtful comments and suggestions. We have made edits to the manuscript based on this review, which are summarized below.
Reviewer 1:
In the present review, Tropea and Chen-Plotkin raise the question whether Parkinson’s disease (PD) patients would be the ideal preclinical population to test Alzheimer’s disease (AD) medication. In other words, they propose that individuals with PD may be even better suited to evaluate the efficiency of these drugs then AD patients themselves. After giving a brief introduction providing background information on both diseases, the authors list features shared between both neurodegenerative disorders and point out that PD patients are at a very high risk to develop concomitant AD. To undermine their claim, the authors summarize several neuropathological, mechanistic and biomarker studies. In the end, they conclude that, as hypothesized in the title, individuals with PD identified with a high risk for AD probably would majorly benefit from receiving early AD treatment and therefore, would also be an easily assessable cohort to evaluate AD-directed medications.
Broad comments
First of all, the review perfectly fits into the scope of the “Journal of Personalized Medicine”, particularly in the special issue “Personalized Medicine for Parkinson's Disease: New Concepts and Future of Individualized Management”, as it discusses how precision medicine can help to identify PD patients at risk for developing concomitant AD and how this population simultaneously could be a well-suited cohort to test AD medications. Secondly, the manuscript is well-structured, well-written and provides concise background information about both neurodegenerative disorders, PD and AD, thus making it easy for the reader to follow the proposed concept. Moreover, the authors clearly lay out the arguments in support of their hypothesis, which are backed up by findings in recent studies. There are only some minor weak points to consider which are listed in the following.
- The wording of the title (“Is PD the ideal preclinical population for AD therapeutics”) is not quite correct as PD itself can’t be a population. Therefore, it would be more precise to write “Are PD patients the ideal preclinical population for AD therapeutics?”
RESPONSE: We appreciate this suggestion and have updated the title as you suggest.
- In the Table, clinical trials testing various AD-directed drugs are listed. It would be interesting to know whether patients with PD diagnosis were excluded from these study cohorts.
RESPONSE: To our knowledge, these trials enrolled participants based on clinical syndrome and in most cases based on biomarker or neuroimaging evidence of AD pathology. It is unlikely that these studies enrolled any subjects with a clinical diagnosis of PD. However, neurodegenerative pathologies commonly co-occur making it likely that people with synuclein pathology but without parkinsonism were enrolled in these studies.
- Minor language recommendations are listed in more detail in the specific comments
Specific comments
1) Language:
- Line 23: insert “the” before “cognitive course”
- Line 29: add “also” before “often”
- Line 56: emphasize this important point by writing“One of the reasons for these disappointing results might be that …”
- Line 137: insert “for” before “dementia”
- Line 174: replace “that” with “than”
- Line 192: it would be clearer to write “…pathology to warrant targeted intervention”
2) Abbreviations:
- Line 98: the abbreviation CSF appears here for the first time, however, it is introduced as late as in line 118
- Line 192: please explain the abbreviation “DRS”
- Line 164: write “NFL” instead of “NfL” to be consistent with nomenclature
- Line 170: please explain “NACC”
3) Figure Legend:
Line 214: add (“right”) after “middle frontal cortex” to be precise about what is shown in the figure
RESPONSE: We appreciate these comments and have made these suggested changes.
We thank you for your helpful feedback. We think the changes we have implemented have strengthened this manuscript.
Reviewer 2 Report
The topis is very interesting and the relationship between the two diseases in many aspects such as neuropathology, neuroradiology and clinical presentations should be deepened.
Author Response
Response to Reviewers re: Manuscript ID: jpm-1305508:
Reviewer 2:
We would like to thank the reviewer for their thoughtful comments and suggestions. We have made edits to the manuscript based on this review, which are summarized below.
The topic is very interesting and the relationship between the two diseases in many aspects such as neuropathology, neuroradiology and clinical presentations should be deepened.
We agree that this is a very interesting topic. Throughout this manuscript we have outlined the relationship between AD and PD to support the premise that PD patients are the ideal population to study AD therapeutics. We have bolstered the sections on neuropathology, neuroradiology, and clinical presentation with additional information and references.
We thank you for your helpful feedback. We think the changes we have implemented have strengthened this manuscript.